# Optimal Extraction of *Ocimum basilicum* Essential Oil by Association of Ultrasound and Hydrodistillation and Its Potential as a Biopesticide Against a Major Stored Grains Pest

**DOI:** 10.3390/molecules25122781

**Published:** 2020-06-16

**Authors:** Eridiane da Silva Moura, Lêda Rita D’Antonino Faroni, Fernanda Fernandes Heleno, Alessandra Aparecida Zinato Rodrigues, Lucas Henrique Figueiredo Prates, Maria Eliana Lopes Ribeiro de Queiroz

**Affiliations:** 1Department of Agricultural Engineering, Universidade Federal de Viçosa, Viçosa 36570-900, Minas Gerais, Brazil; annne.moura@gmail.com (E.d.S.M.); fernandafhg@gmail.com (F.F.H.); azinato@yahoo.com.br (A.A.Z.R.); hlucash@gmail.com (L.H.F.P.); 2Department of Chemistry, Universidade Federal de Viçosa, Viçosa 36570-900, Minas Gerais, Brazil; meliana@ufv.br

**Keywords:** *Sitophilus zeamais*, yield, drying temperature, linalool, estragole, chromatographic analysis

## Abstract

The essential oil of basil (*Ocimum basilicum*) has significant biological activity against insect pests and can be extracted through various techniques. This work aimed to optimize and validate the extraction process of the essential oil of *O. basilicum* submitted to different drying temperatures of the leaves and extracted by the combination of a Clevenger method and ultrasound. The biological activity of the extracted oil under different conditions was evaluated for potential control of *Sitophilus zeamais*. The extraction method was optimized according to the sonication time by ultrasound (0, 8, 19, 31 and 38 min) and hydrodistillation (20, 30, 45, 60 and 70 min) and drying temperature (20, 30, 45, 60 and 70 °C). The bioactivity of the essential oil was assessed against adults of *S. zeamais* and the effects of each variable and its interactions on the mortality of the insects were evaluated. The best yield of essential oil was obtained with the longest sonication and hydrodistillation times and the lowest drying temperature of leaves. Higher toxicity of the essential oil against *S. zeamais* was obtained by the use of ultrasound for its extraction. The identification and the relative percentage of the compounds of the essential oil were performed with a gas chromatograph coupled to a mass selective detector. The performance of the method was assessed by studying selectivity, linearity, limits of detection (LOD) and quantification (LOQ), precision and accuracy. The LOD and LOQ values for linalool were 2.19 and 6.62 µg mL^−1^ and for estragole 2.001 and 6.063 µg mL^−1^, respectively. The coefficients of determination (*R*^2^) were >0.99. The average recoveries ranged between 71 and 106%, with coefficient of variation ≤6.4%.

## 1. Introduction

Basil (*Ocimum basilicum*) is one of the most widely used aromatic plants. Due to its chemical composition, it is used in perfumery, cooking, food industry, toothpaste, in the production of cosmetics and medicine [1]. Currently, the essential oil of *O. basilicum* has been studied as an alternative in the control of stored grain pests [2]. The essential oil of *O. basilicum* exhibit a wide variety of chemical compounds, depending on the variations of the chemotypes, leaf, flowers and plant origin [3]. Its main components include estragole, linalool and eugenol [4].

Considering that the quality of the oil is related to its chemical composition, the choice of an adequate extraction method is extremely important [5]. The employed method should not influence the composition and avoiding the decomposition of compounds of interest, in order to maintain the quality of the final product [6,7].

Several techniques are used for the extraction of essential oils, such as: classic distillation of water vapor, percolation, extraction of solid fluid and the supercritical fluid extraction method [8]. Steam distillation (SD) is presented in one of the most conventional methods, hydrodistillation (HD), where the vapor evolves from the herb’s suspension in the balloon. Hydrodistillation is carried out using the Clevenger apparatus [9]. Despite of being the most widely used method, it has some important deficiencies, such as the difficulty of controlling heat transfer in a constant manner throughout the process and extensive time of extraction. Thus, resulting in higher operational costs and can induce the hydrolysis of some essential oil constituents. All of that might reduce yield and cause losses of volatile compounds due to prolonged heating [10].

More modern versions of the essential oil extraction system have been used. Microwave heating can be used for distillation or pre-treatment [11,12]. As pre-treatment, it is possible to use enzymatic digestion, where different fiber breakdowns (cellulase, hemicellulase, xylanase), pectin degradation (pectinase) or other enzymes (such as protease) are used [13,14]. Microwave-assisted hydrodistillation (MHD) usually involves the use of freshly harvested parts of plants, since the required water vapor is provided by the natural moisture of the plant.

Ultrasonication is another promising technique that has been applied in the pre-treatment of plant parts. It is applied to avoid the use of solvents and reduce the processing time for the extraction of essential oils [5]. The technique has also been applied to improve the extraction of polysaccharides and essential oils from plant material, mainly through the phenomenon of cavitation [15,16]. The mechanical effect of ultrasound might accelerate the release of organic compounds contained in the plant’s body due to cell action, wall rupture, intensification of mass transfer and greater ease of solvent access to cell content [16].

The use of ultrasound increases the purity of extracted components in less time and at lower temperatures [17]. Its effectiveness in intensifying the essential oil extraction process has been already proved [18,19,20], as well as in the adaptation of conventional devices [21].

In view given the importance of the extraction process in the yield and quality of essential oils, the objective of this work was to optimize and validate the extraction process of the essential oil of *O. basilicum* submitted to different drying temperatures of the leaves and extracted by the combination of Clevenger method and ultrasound. The optimization aimed to increase the yield of the extracted oil and maintain its quality. Considering the possible application as biopesticides, the toxicity of the essential oil was evaluated to adults of *Sitophilus zeamais.*

## 2. Results

### 2.1. Optimization of the Extraction Process

#### 2.1.1. Yield of Essential Oil

The Pareto graph with data of the yield of the essential oil of *O. basilicum* leaves in relation to dry matter when subjected to drying at different temperatures of the leaves, different hydrodistillation, and ultrasound times is shown in Figure 1, respectively. The results of the factorial design show that the investigated variables (drying temperature, ultrasound time and hydrodistillation time) were significant (*p* ≤ 0.05) for the yield of *O. basilicum* essential oil (Figure 1, Figure 2 and Figure 3).

According to Figure 1, the higher yields of essential oil were proportional to the increase in drying temperature, hydrodistillation time and ultrasound time. The highest yield of essential oil and greater toxicity on adults of *S. zeamais* were obtained in treatments 7, 12, 14, 25 and 27 (Table 1) with ultrasound times of 19, 31 and 38 min, hydrodistillation time of 45, 60 and 70 min and drying temperature 30, 45, 60 and 70 °C. The lowest yield of essential oil was obtained in treatments 9 and 26, in which there was no use of ultrasound in the extraction, which shows that the longer the exposure time to ultrasound and hydrodistillation, the greater the yield of essential oil. The yield of the EO of *O. basilicum* also increased with the increase in the drying temperature of its leaves (Figure 3).

#### 2.1.2. Toxicity to *S. zeamais*

Results of the central composite planning show that the variables investigated (drying temperature, time of sonication by ultrasound and hydrodistillation time) were significant (*p* ≤ 0.05) for the toxicity of the essential oil of *O. basilicum* to *S. zeamais* (Figure 4). Linear effects for sonication time, hydrodistillation time, drying temperature, and the linear interaction of sonication time and drying temperature were signigicant. On the other hand, quadratic effects for sonication time and and hydrodistillation were also significant.

The mortality of *S. zeamais* was higher when the essential oil was extracted with a longer time of ultrasound and hydrodistillation (Figure 5). Which means that in addition to increasing the yield of EO, such extraction conditions increase its toxicity to adults of *S. zeamais*.

### 2.2. Major Component Identification

The chemical composition of the essential oil of the leaves of *O. basilicum* was carried out using GC-MS. The main constituents were identified by comparison of the mass spectrum of the compounds with the NIST 14 library spectra and by calculating the Kovats index for a series of saturated alkanes (C_7_–C_30_) (49451-U, 99.0%, Supelco, Bellefonte, PA, USA). Chromatographic analysis showed the presence of two major constituents in the basil EO (Table 2). Estragole is the major compound, representing 85% of the identified compounds, followed by linalool at 12%, respectively (Figure 6).

#### Quantification of Major Components

The optimized association of ultrasound and hydrodistillation was applied to quantify linalool and estragole in samples of essential oil of *O. basilicum* by GC-FID. The five extraction conditions were used which provided a higher yield of essential oil associated with greater toxicity of insects.

The concentration of each compound was calculated based on their chromatographic area and the analytical curve of each compound. The highest amount of linalool (12.2 mg mL^−1^) and estragole (51.6 mg mL^−1^) was obtained in the samples extracted with an ultrasound time of 31 min, 60 min of hydrodistillation and drying temperature of leaves of 30 °C (Table 3). These conditions of extraction of the compounds were applied in the method validation process.

### 2.3. Method Validation

#### 2.3.1. Selectivity

To assess selectivity, the optimized method was applied to samples of linalool and estragole. The chromatograms for these samples are shown in Figure 7. The comparison of the two chromatograms shows that there are no coincident peaks between the original sample of toluene and the sample with the compounds, thus the analytes of interest are not affected by the analytes of the original sample, which proves the good selectivity of the proposed method.

#### 2.3.2. Linearity and Limits of Detection and Quantification

The analytical curve is described by the relationship between the theoretical concentration of the analyte in the sample and the chromatographic area. Linearity was evaluated by analyzing samples of linalool and estragole in different concentrations (0.5; 5; 10; 20; 30; 50 and 60 mg mL^−1^ for linalool) (0.25; 0.5; 1; 5; 10; 50 and 100 mg mL^−1^ for estragole) and subjected to the extraction and analysis method. The data were used to construct analytical curves (*n* = 7, analyzed in triplicates) relating the chromatographic area of the analytes versus their concentrations. The determination coefficient (*R*^2^) obtained for both compounds were >0.99, which indicates the good linearity of the method for the concentration range evaluated (Figure 8A,B).

The limits of detection (LOD) and quantification (LOQ) of the method were calculated as the concentration giving a signal-to-noise ratio of three (S/N = 3) and ten (S/N = 10), respectively. In this study, the LOD and LOQ obtained for linalool was 2.19 and 6.62 µg mL^−1^ and for estragole was 2.001 and 6.063 µg mL^−1^, respectively.

#### 2.3.3. Precision and Accuracy

Precision and accuracy were assessed at the levels of 14 µg mL^−1^ (2 × LOQ), 35 µg mL^−1^ (5 × LOQ) and 70 µg mL^−1^ (10 × LOQ) in seven replicates (Table 4). The method showed good extraction efficiency and analytical frequency at the concentrations used with a coefficient of variation ≤6.4%. For the three concentration levels evaluated, the average recovery ranged from 71 to 77% for linalool and of 74 to 106% for estragole. Recoveries between 70 and 120% and coefficients of variation lower than 20% for analytes are considered satisfactory, indicating the accuracy and precision of the method [23].

## 3. Discussion

The increase in yield of EO of *O. basilicum* after using ultrasound in the extraction process might be due to the cavitation generated by sound waves, releasing heat, causing the rupture of plant cells and, therefore, increasing the extraction efficiency [24,25]. Similarly, the yield of *Salvia* essential oil also increased after its extraction with the use of ultrasound sonication [26]. Considering the low mechanical resistance of the cell membranes of the glands that contain essential oils, it is possible to extract them efficiently in a short period of time with the use of ultrasound [19].

The use of ultrasound as a pre-treatment, for example, allows a reduction of about 70% in the extraction time in relation to conventional hydrodistillation. In addition, these conditions allow an increase in the extraction of bioactive compounds and consequently improve the antioxidant and antimicrobial activity of the essential oils obtained [5]. Using ultrasound to extract the essential oil of *Mentha piperita* L., *Origanum majorana* L., and *Chamomilla recutita* L. increased the yield by 12% [27].

The yield of EO of *O. basilicum* increased with the increase in the drying temperature of its leaves. Similar results were obtained for lemongrass essential oil, whose yield increased with an increase in the drying temperature [28]. The effect of drying temperature on the yield of the essential oil of *Tanaecium nocturnum* was evaluated by Pimentel et al. and the authors concluded that the increase on drying temperature (30–40 °C) did not affect the oil content, but caused losses and variations in the amounts of its compounds [29].

The longer time of ultrasound and hydrodistillation in the extraction of basil essential oil caused greater mortality in *S. zeamais*, which means that in addition to increasing the yield of EO, such extraction conditions increased its toxicity to adults of *S. zeamais*. The optimized extraction conditions provided higher yield and also higher bioactivity of the essential oil against a major insect pest of stored grains. This may be due to the use of ultrasound that increased the extraction of bioactive compounds and, consequently, improved the biological activity of the essential oil obtained [5]. As observed in the literature, the chirality of linalool may influence its biological activity. Although not assessed in the present study, the enantiomeric ratio of chiral compounds might account for the differences observed in the biological activity of the essential oil extracted under different conditions against *S. zeamais* [30,31,32].

Essential oils have a wide application in perfume and flavouring industries and have been studied for their application in pest management, for example, as biopesticides [33,34,35]. The bioactivity of essential oils in pest management of stored grains is linked not only to their lethality, but also to behavioural changes aiming the increase in food safety and quality [33,36,37,38]. Together with other factors, like the diffusion of volatile components [39,40] and the persistence of components after application [41], the compositition of essential oils is important to predict its bioactivity.

The chromatographic analysis of the basil EO shows that estragole and linalool are the major compounds of the oil. While estragole represents 85% of the identified compounds, linalool accounts for 12% of the identified compounds. Similar results were found by [42,43,44,45]. However, the composition of essential oils has great variability in terms of chemical and biological aspects, depending on the climatic condition, location, genetic variability, sections of the plant used for extractions and harvesting time of the plant material [46,47]. Genetic and environmental factors, as well as plant ontogenesis, also have a determining effect on the composition and quality of the essential oil of *O. basilicum* [45].

The validated analytical method herein proposed for analysis of the essential oil showed good selectivity. Selectivity is used to ensure that the peak response of the analyte (assessed at the characteristic retention time) comes from the analyte and not from other compounds present in the sample [48]. The proposoed method also showed good linearity for both components in the evaluated concentration ranges (*R*^2^ > 0.99). The linearity of a method’s response corresponds to its ability to demonstrate that the results obtained (chromatographic areas) are directly proportional to the concentration of the analyte in the sample within a specific range [49].

The highest amount of linalool and estragole was obtained in the samples extracted with an ultrasound time of 31 min, 60 min of hydrodistillation and a drying temperature of 30 °C. Generally low drying temperatures (up to 30 °C) tend to preserve the chemical composition of essential oils since high temperatures can cause losses and chemical changes in their volatile compounds [50]. As observed in the literature, the increase in the drying temperature (40, 50, 60 and 70 °C) caused a decrease in the relative proportion of monoterpenes and sesquiterpenes of the essential oil of *Ocimum selloi* [51].

According to Randuz et al. the variability on compounds concentration can be caused by oxidation and reduction reactions, and rearrangements caused during the drying process at high temperatures [48]. In samples submitted to drying at 70 °C, it was observed that all the constituents of the EO of *O. basilicum* suffered reductions in their levels, indicating possible losses due to volatilization [52].

The higher quality of essential oil extracted with the use of ultrasound can be attributed to lower degradation of thermosensitive compounds [19,27,53]. In addition, changes in composition are related to facilitated release of essential oils from secretory glands or to transformations of unstable chemical compounds during ultrasound application [54,55]. The use of ultrasound in the pre-treatment to extract essential oil from lavender flowers resulted in a 10% increase in the concentration of linalool [20]. In the extraction of essential oil from *Daucus carota*, the use of ultrasound in the pre-treatment promoted an increase in the concentration of monoterpenes and sesquiterpenes [55]. Altogether, it’s shown that the use of ultrasound affects the chemical composition of essential oils.

## 4. Material and Methods

### 4.1. Reagents

Toluene HPLC-grade solvent (Sigma-Aldrich, 99.9%, Darmstadt, Germany) was used to dillute linalool and estragole analytical standards (Sigma-Aldrich, 99.9%). Standard solution of alkanes (C_7_–C_30_) (49451-U, Supelco, 99.0%), was also purchased from Sigma-Aldrich. Working solutions at the appropriate concentrations were prepared directly from the stock solutions of the standards and used for the construction of analytical curves of linalool and estragole. All standard solutions were stored at −18 °C.

### 4.2. Samples

Samples of *O. basilicum*, produced without the use of pesticides, were purchased directly from a producer in Viçosa (Minas Gerais, Brazil). Immediately after harvesting, the samples were packed in polyethylene bags and taken to the Post-Harvest Laboratories of the Agricultural Engineering Department of the Universidade Federal de Viçosa (UFV), where the drying of fresh leaves and the extraction experiments of *O. basilicum* essential oil were conducted.

### 4.3. Samples Preparation

A greenhouse with forced air circulation (Didática SP 1152 L, São Paulo, Brazil) was used for drying *O. basilicum* leaves. Six perforated trays with 100 g of fresh leaves each were used. During the drying process, the trays were weighed at intervals of 30 min on a semi-analytical scale (Gehaka, São Paulo, Brazil, model BG 4400, with accuracy of ±0.01/0.1 g), in order to control the drying until they reached constant weight. Five drying temperatures of leaves (20, 30, 45, 60 and 70 °C) were used and the experiment was conducted using factorial design (Table 5).

### 4.4. Optimization of Essential Oil Extraction

For the extraction of the essential oil of *O. basilicum*, a new method was developed based on the association of hydrodistillation and ultrasound techniques. The extractions were carried out by hydrodistillation using a Clevenger-type apparatus with samples of 20 g of hand-grounded leaves dried under different temperatures. The crushed leaves were placed in a 250 mL flask. The balloon was attached to Clevenger and placed in an ultrasound device (Cristófoli, Paraná, Brazil), where it was submerged in water at 100 °C and subjected to different sonication (0, 8, 19, 31 and 38 min) and hydrodistillation (20, 30, 45, 60 and 70 min) times (Figure 9). At the end of this period, the essential oil was collected and separated from the hydrolate by freezing. The extracted oil was quantified and its yield was calculated.

### 4.5. Factorial Planning

The extraction of essential oil was optimized in order to reduce the extraction time and maintain the quality of the oil using a central composite design with six repetitions at the central point (Table 1). Three variables were studied: ultrasound sonication time (UST), hydrodistillation time (HDT) and leaves drying temperature (DT). The variables were studied at two levels and the analyzes were performed in triplicate. The effects of each variable and the interactions between variables on oil yield and mortality of *S. zeamais*, were calculated using the software Statistica 12.0 (StatSoft Corp., Tulsa, OK, USA). The data were presented in graphs generated using the SigmaPlot 12.5 software (Systat Software, San Jose, CA, USA).

### 4.6. Yield of Essential Oil

The percentage of yield calculation was performed following the method recommended by the Instituto Adolfo Lutz [56] (Equation (1)):Essential oil content = (V/m) × 100(1)
where: V = volume of essential oil distilled (mL) and m = initial weight of plant material (g).

The data were submitted to analysis of variance (ANOVA) and their means compared by Tukey’s test at 5% probability, using the Statistica 12.0 software. Samples of the essential oil of *O. basilicum* that obtained the highest yield according to the extraction conditions and greater toxicity on adults of *S. zeamais* (Table 1) were used to identify and quantify the main components of the oil and to validate the extraction method.

### 4.7. Fumigation Toxicity Bioassay

The bioassays were performed in 0.8 L glass flasks (8 cm in diameter × 15 cm in height) with 50 adults of non-sexed *S. zeamais*, in four replicates. The concentration of essential oil used was 20 μL L^−1^ of air. The working solutions with samples of the the essential oil were prepared using toluene as solvent and applied with a microsyringe (Hamilton, Reno, NV, USA) on filter-paper discs with a diameter of 4.4 cm placed in Petri dishes (diameter 6.5 cm). The Petri dishes were covered with organza fabric and placed at the bottom of the flasks. 25 μL of pure solvent (toluene) was used as a control. After the distribution of the insects, the flasks were closed with a screw-on metallic cap and sealed with parafilm PM996 (American, Neenah, USA) to avoid leakage of volatiles from the essential oil during the exposure period. The flasks were kept for 24 h in a B.O.D. (model 347 CD, Fanem, São Paulo, Brazil) at a temperature of 27 ± 2 °C. After this period, dead and live insects were counted. The mortality of *S. zeamais* exposed to the essential oil was corrected in relation to the mortality of insects exposed to the solvent using Abbott’s formula [57].

### 4.8. Majority Component Identification by GC-MS

The identification and the relative percentage of the compounds of the essential oil were carried out at the Chemistry Department of the of Universidade Federal de Viçosa (UFV), Brazil. The essential oil of *O. basilicum* was analyzed by gas chromatography coupled to mass spectrometry (GC-MS) using a QP2010 system (Shimadzu, Japan) under the following conditions: capillary column of fused silica (30 m length and 0.25 mm of internal diameter) with RTX^®^-5MS stationary phase (0.25 µm film thickness) and helium as carrier gas with a flow rate of 1.0 mL/min. Injector temperature of 220 °C and initial temperature of the column as 60 °C. Heating rate was set to increase 2 °C min^−1^ up to 200 °C, and 5 °C min^−1^ up to 250 °C. The mass spectra were obtained by electron impact at 70 eV, with a scan from 29 to 400 (*m*/*z*). Samples of the extracted oil were diluted in toluene to the concentration of 10 mg mL^−1^ and 1 µL was injected with a split ratio of 1:20. The total analysis time was 80 min. The identification of the compounds was carried out by comparing the mass spectra in the NIST library, visual interpretation of the mass spectra and confirmed by Kovats Index (KI) calculation and comparison with the literature [22]. The KI of each compound was calculated based on the retention time of the compounds and the alkanes of the standard solution of alkanes. The relative percentage of each compound was calculated using the ratio between the area of each peak and the total area of all constituents in the sample.

### 4.9. Quantification of Major Components

A GC2010 gas chromatograph (Shimadzu) equipped with a flame ionization detector (GC-FID) was used for the quantification of the main components of samples of *O. Basilicum* essential oil. The best extraction conditions obtained from factorial planning were evaluated by comparing the peak area values (chromatographic responses) for the analytes of interest in each assay. Chromatographic separation was performed on a DB-5 capillary column (Agilent Technologies, Palo Alto, CA, USA), with a stationary phase composed of 5% phenyl and 95% dimethylsiloxane (30 m × 0.25 mm, 0.10 µm film thickness). Nitrogen (99.999%, Air Products, São Paulo, Brazil) was used as carrier gas in a flow of 1.82 mL min^−1^ and a split ratio of 1:5. Injector and detector temperatures were set at 220 and 300 °C, respectively. The initial column temperature was 60 °C with heating rate at 7 °C min^−1^ up to 120 °C and maintained at this temperature for 1 min. The total time of analysis was 8.5 min. Five solutions of essential oil were injected at a concentration of 10 mg mL^−1^ in toluene, in three repetitions. Chromatographic parameters were managed using CGsolution software (Shimadzu, Kyoto, Japan). The compounds were identified by comparing the retention times of the peaks obtained for samples of the essential oil and those of the standards. The concentration of each compound was calculated based on their chromatographic area and the analytical curve of each compound.

### 4.10. Validation of the Method

The optimized method was validated using the best conditions for the extraction of the main components from the essential oil of *O. basilicum.* The validation of the analysis of linalool and estragole extracted by combined ultrasound and hydrodistillation techniques and their analysis by (GC-FID) was based on the following parameters: selectivity, linearity, limit of detection (LOD), limit of quantification (LOQ), accuracy (recovery assays), and precision (repeatability).

The validation of the method was developed according to the standards of the guide [25]. Solutions of both compounds in concentrations ranging from 0.25 to 100 mg mL^−1^ were used to construct the analytical curve of linalool and estragole. The limits of detection and quantification of the analytical curve of linalool and estragole were determined by means of signal-to-noise.

### 4.11. Statistical Analysis

Data related to the yield of essential oil and mortality of *S. zeamais* were subjected to analysis of variance (ANOVA) and regression analysis by surface response as a function of hydrodistillation time, ultrasound time and drying temperature. The analytical curve for quantification of linalool and estragole were subjected to ANOVA and regression analysis. The graphs were plotted using the SigmaPlot 12.5 software (Systat Software, San Jose, CA, USA). All statistical tests were performed considering a confidence level of 95% by Statistica 12.0 software (StatSoft Corp., Tulsa, OK, USA).

## 5. Conclusions

The method of extracting the essential oil of *O. basilicum* by the association of ultrasound and hydrodistillation techniques was developed and validated. In general, the highest amount of linalool and estragole was obtained in the samples extracted with an ultrasound time of 31 min, 60 min of hydrodistillation and drying temperature of leaves of 30 °C. The bioactivity of the essential oil was assessed by its toxicity to *S. zeamais*, which increased with the use of ultrasound in the extraction process.

The validation of the optimized method assessed for its selectivity, limits of detection and quantification, linearity, accuracy and precision showed that the proposed method is sensitive, precise and accurate (mean recoveries were between 71 and 106%), with coefficient of variation ≤6.4%. Values of LOD and LOQ for linalool were 2.19 and 6.62 µg mL^−1^ and for estragole 2.001 and 6.063 µg mL^−1^, respectively. Clevenger method and ultrasound were, for the first time, combined into a method for the extraction of essential oil of *O. basilicum*.

## Figures and Tables

**Figure 1 molecules-25-02781-f001:**
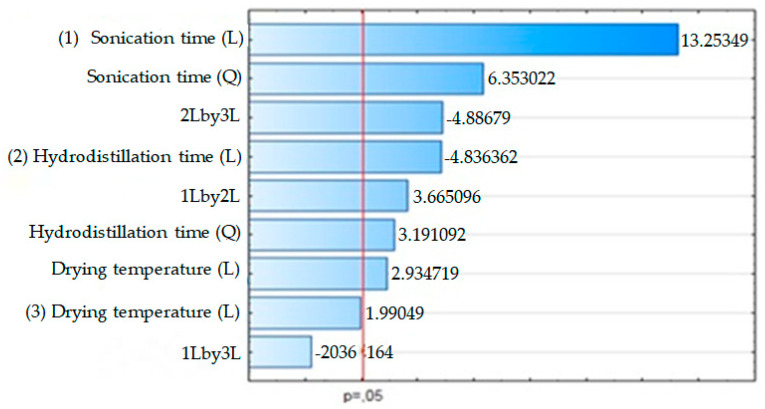
Pareto graph of standardized effects for the performance variables of *O. basilicum* essential oil as a function of drying temperature, hydrodistillation time and ultrasound time.

**Figure 2 molecules-25-02781-f002:**
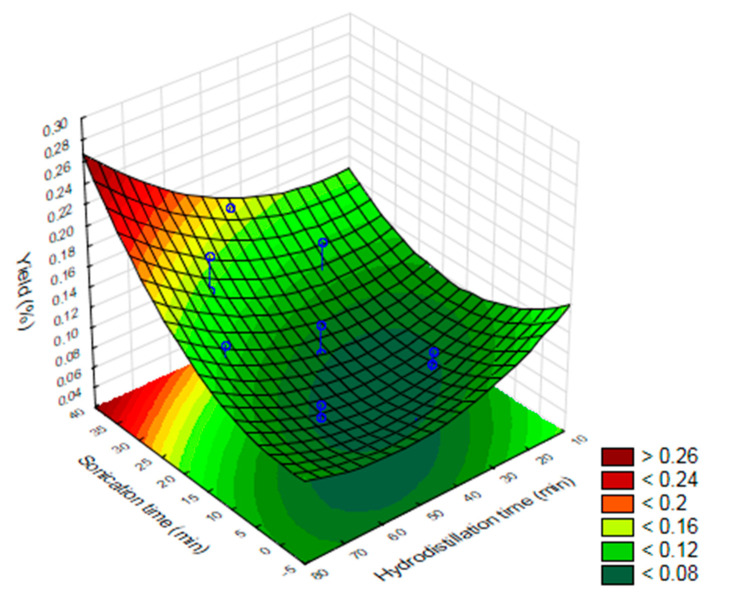
Response surface of the optimization of the extraction of the essential oil of *O. basilicum* by the Clevenger method associated with ultrasound as a function of hydrodistillation time and ultrasound time.

**Figure 3 molecules-25-02781-f003:**
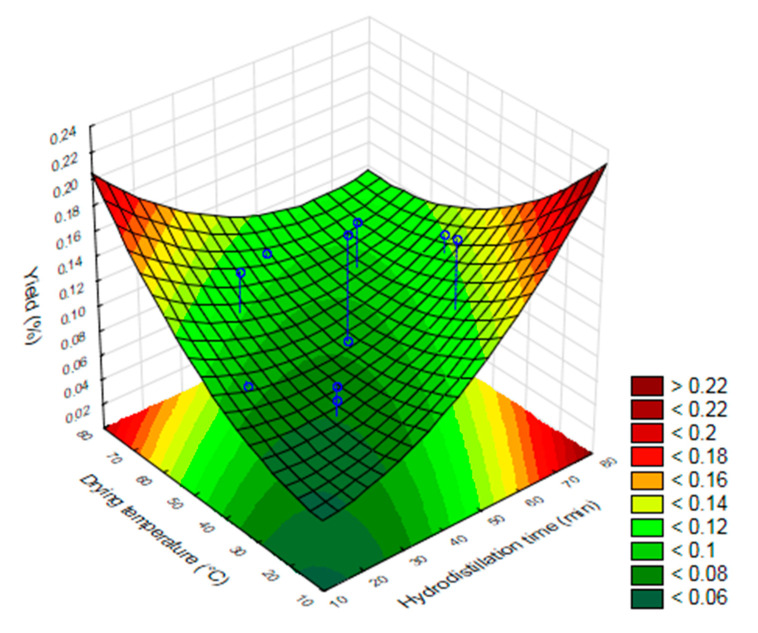
Response surface of the optimization of the extraction of the essential oil of *O. basilicum* by the clevenger method associated with the ultrasound as a function of the hydrodistillation time and drying temperature.

**Figure 4 molecules-25-02781-f004:**
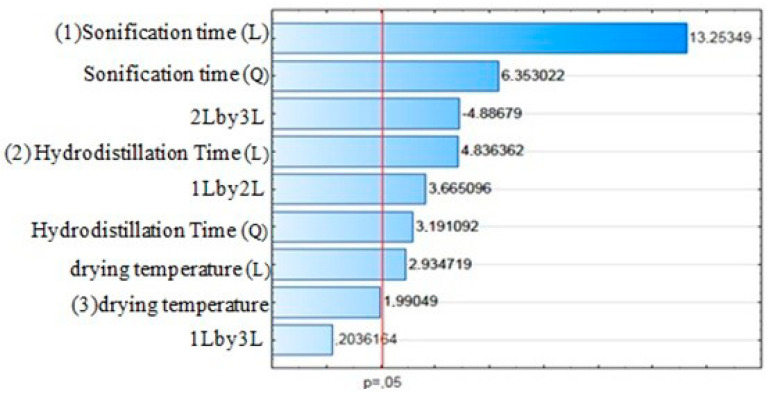
Pareto chart of standardized effects for the mortality variable of *S. zeamais* by the essential of *O. Basilicum* as a function of drying temperature, hydrodistillation time and ultrasound time.

**Figure 5 molecules-25-02781-f005:**
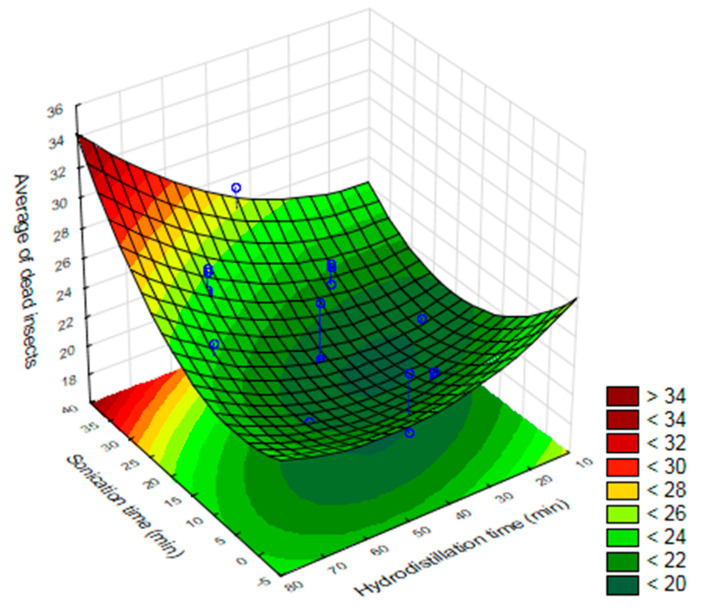
Response surface of the optimization of the extraction of the essential oil of *O. basilicum* for the mortality variable of *S. zeamais* as a function of hydrodistillation time and ultrasound time.

**Figure 6 molecules-25-02781-f006:**
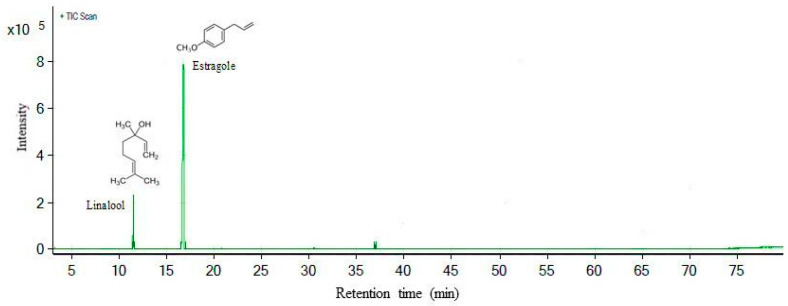
Chromatogram of the identification of the main components of the essential oil of *O. basilicum* by GC-MS. where: retention time of 11.50 min corresponds to linalool and retention time of 16.75 min corresponds to estragole.

**Figure 7 molecules-25-02781-f007:**
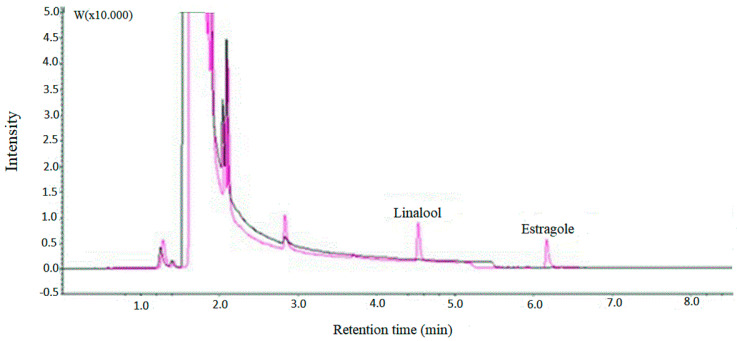
Chromatogram obtained from a pure toluene sample (black), followed by the chromatogram (pink) obtained from the sample containing linalool (retention time 4.57 min) and estragole (retention time 6.38 min) after extraction and analysis by GC-FID.

**Figure 8 molecules-25-02781-f008:**
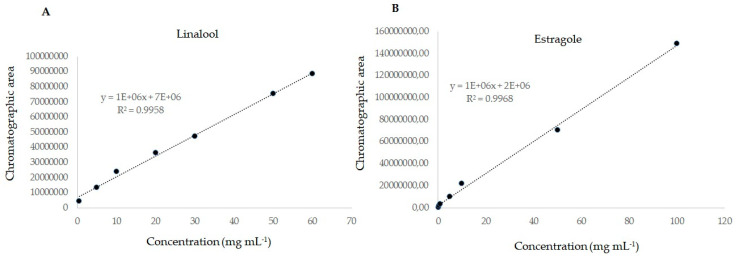
Analytical curve of the chromatographic response as a function of the concentration of linalool (**A**) and estragole (**B**) in mg mL^−1^ of toluene.

**Figure 9 molecules-25-02781-f009:**
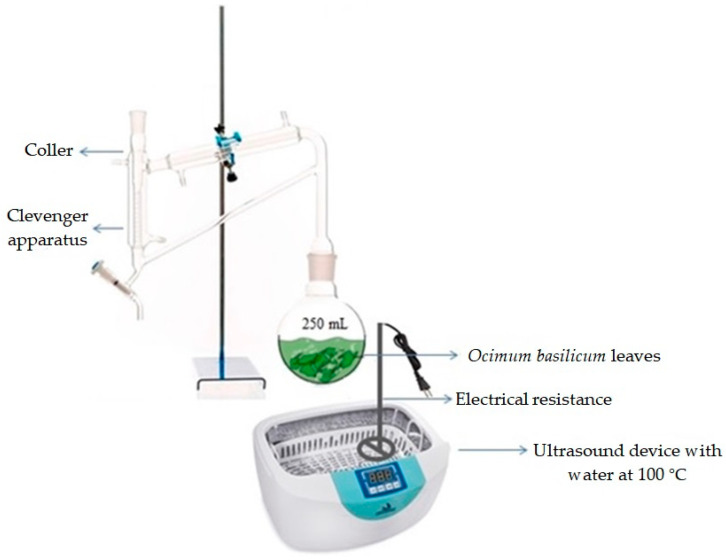
Scheme of the method used to extract the essential oil of *O. basilicum*.

**Table 1 molecules-25-02781-t001:** Effect of ultrasound time, hydrodistillation time and drying temperature on the yield of *O. basilicum* essential oil and its toxicity on adults of *S. zeamais*.

Experiment	Replicate	Sonication Time (min)	HD Time (min)	Drying Temperature (°C)	Mortality (Average of Dead Insects)	Average Oil Yield (%)
1	1	8 (‒)	30 (‒)	30 (‒)	20	0.08
2	1	8 (‒)	30 (‒)	30 (‒)	21	0.09
3	1	8 (‒)	60 (+)	60 (+)	20	0.09
4	1	8 (‒)	60 (+)	60 (+)	21	0.08
5	1	31(+)	30 (‒)	30 (‒)	21	0.09
6	1	31 (+)	30 (‒)	60 (+)	23	0.13
7 *	1	31(+)	60 (+)	30 (‒)	24	0.16
8	1	31(+)	60 (+)	60 (+)	25	0.13
9	1	0 (0)	45 (0)	45 (0)	20	0.07
10	1	38 (+α)	45 (0)	45 (0)	25	0.16
11	1	19 (0)	20 (-α)	45 (0)	21	0.08
12 *	1	19 (0)	70 (+α)	45 (0)	25	0.09
13	1	19 (0)	45 (0)	20 (-α)	20	0.08
14 *	1	19 (0)	45 (0)	70 (+α)	24	0.11
15	1	19 (0)	45 (0)	45 (0)	20	0.08
16	1	19 (0)	45 (0)	45 (0)	20	0.08
17	1	19 (0)	45 (0)	45 (0)	21	0.08
18	2	8 (‒)	30 (‒)	30 (‒)	20	0.08
19	2	8 (‒)	60 (+)	30 (‒)	21	0.09
20	2	8 (‒)	30 (‒)	60 (+)	20	0.09
21	2	8 (‒)	60 (+)	60 (+)	21	0.08
22	2	31 (+)	30 (‒)	30 (‒)	21	0.09
23	2	31 (+)	60 (+)	60 (+)	23	0.13
24	2	31 (+)	30 (‒)	30 (‒)	23	0.16
25 *	2	31 (+)	60 (+)	60 (+)	26	0.13
26	2	0 (-α)	45 (0)	45 (0)	24	0.06
27 *	2	38 (+α)	45 (0)	45 (0)	28	0.16
28	2	19 (0)	45 (0)	45 (0)	20	0.08
29	2	19 (0)	45 (0)	45 (0)	24	0.13
30	2	19 (0)	45 (0)	20 (-α)	20	0.08
31	2	19 (0)	70 (+α)	70 (+α)	24	0.11
32	2	19 (0)	45 (0)	45 (0)	20	0.08
33	2	19 (0)	45 (0)	45 (0)	20	0.08
34	2	19 (0)	45 (0)	45 (0)	21	0.08

HD—Hydrodistillation. * Experiments highlighted (samples of the essential oil of *O. basilicum* that obtained the highest yield and greater toxicity on adults of *S. zeamais*).

**Table 2 molecules-25-02781-t002:** Chemical composition and relative percentage of the compounds identified in the essential oil of *O. basilicum* using gas chromatography analysis coupled with mass spectrometry (GC-MS).

Constituent	^a^ RI Literature	^b^ RI Calculated	Relative Percentage (%)
Estragole	1195	1169	85
Linalool	1095	1071	12

^a^ Relative retention index taken from [22] and/or NIST 14. ^b^ Retention index experimentally determined using homologous series of C_7_–C_30_ alkanes (Kovats index).

**Table 3 molecules-25-02781-t003:** Quantification of linalool and estragole from samples of essential oil of *O. basilicum* that obtained the highest yield through extraction by the association of hydrodistillation and ultrasound techniques.

Extraction Conditions	Linalool (mg mL^−1^)	Estragole (mg mL^−1^)
19UST 45HDT 70DT	6.9	31.8
31UST 60HDT 60DT	8.7	37.6
38UST 45HDT 45DT	9.4	39.8
19UST 70HDT 45DT	10.5	47.8
31UST 60HDT 30DT	12.2	51.6

UST—Ultrasound Time (min); HDT—Hydrodistillation Time (min) and DT—Drying Temperature (°C).

**Table 4 molecules-25-02781-t004:** Accuracy and precision of the method of quantification of linalool and estragole by chromatograph equipped with flame ionization detector (GC-FID).

Essential Oil	Accuracy (*n* = 7)			Repeatability (Intra-Day, *n* = 7)		
	Recovery (%)			Coefficient of Variation (%)		
	2 × LOQ	5 × LOQ	10 × LOQ	2 × LOQ	5 × LOQ	10 × LOQ
Linalool Estragole	71.0	75.0	77.0	4.2	6.4	1.0
	74.0	83.0	106.0	0.2	0.1	0.2

*n*—number of replicates; LOQ—limit of quantification.

**Table 5 molecules-25-02781-t005:** Central composite design with six repetitions at the central point (C) to investigate the effect of ultrasonication time, hydrodistillation time and drying temperature on the yield of *O. basilicum* essential oil and its toxicity on adults of *S. zeamais*.

Experiment	Replicate	Sonication Time (min)	Hydrodistillation Time (min)	Leaf Drying Temperature (°C)
1	1	8 (−)	30 (−)	30 (−)
2	1	8 (−)	30 (−)	30 (−)
3	1	8 (−)	60 (+)	60 (+)
4	1	8 (−)	60 (+)	60 (+)
5	1	31(+)	30 (−)	30 (−)
6	1	31 (+)	30 (−)	60 (+)
7	1	31(+)	60 (+)	30 (−)
8	1	31(+)	60 (+)	60 (+)
9	1	0 (0)	45 (0)	45 (0)
10	1	38 (+α)	45 (0)	45 (0)
11	1	19 (0)	20 (−α)	45 (0)
12	1	19 (0)	70 (+α)	45 (0)
13	1	19 (0)	45 (0)	20 (−α)
14	1	19 (0)	45 (0)	70 (+α)
15	1	19 (0)	45 (0)	45 (0)
16	1	19 (0)	45 (0)	45 (0)
17	1	19 (0)	45 (0)	45 (0)
18	2	8 (−)	30 (−)	30 (−)
19	2	8 (−)	60 (+)	30 (−)
20	2	8 (−)	30 (−)	60 (+)
21	2	8 (−)	60 (+)	60 (+)
22	2	31 (+)	30 (−)	30 (−)
23	2	31 (+)	60 (+)	60 (+)
24	2	31 (+)	30 (−)	30 (−)
25	2	31 (+)	60 (+)	60 (+)
26	2	0 (−α)	45 (0)	45 (0)
27	2	38 (+α)	45 (0)	45 (0)
28	2	19 (0)	45 (0)	45 (0)
29	2	19 (0)	45 (0)	45 (0)
30	2	19 (0)	45 (0)	20 (−α)
31	2	19 (0)	70 (+α)	70 (+α)
32	2	19 (0)	45 (0)	45 (0)
33	2	19 (0)	45 (0)	45 (0)
34	2	19 (0)	45 (0)	45 (0)

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
