# Peer review of "Optimal Extraction of Ocimum basilicum Essential Oil by Association of Ultrasound and Hydrodistillation and Its Potential as a Biopesticide Against a Major Stored Grains Pest"

_molecules, 2020, doi:10.3390/molecules25122781_

Round 1

Reviewer 1 Report

Manuscript is well designed, written and discussed; however, there are important aspects which must be corrected prior to its acceptance.

Authors claimed that they are optimizing the extraction of essential oils, however they are also using different drying temperatures of the raw material, so, authors are optimizing two aspects, and therefore it must by shown in the title, introduction and objectives.

This referee does not understand the the objective of the toxicity assay. Clarify this aspect if you consider it relevant on the research, if not eliminate please.

Line 151: there is a reference not properly presented for the Molecules format.

Please use always same way of presentation of GC techniques, GC-MS and GC-FID, do not use GC/FID.

Authors must include another identification system by calculations of retention indexes (perhaps kovats) and compare them with the literature.

Manuscript needs major revisions.

Author Response

Viçosa, May 26, 2020

We are re-presenting the manuscript Manuscript ID molecules-798684 entitled "Optimization and validation of the Ocimum basilicum essential oil extraction method by the association of ultrasound and hydrodistillation techniques" with revision to be considered for publication at Molecules. We thank the editor and reviewers for their comments and suggestions as they have greatly improved the work. The modified sentences were marked in red and the English revisions in green.

Yours sincerely,

 Prof. Dra. Lêda Rita D’Antonino Faroni

Department of Agricultural Engineering

Universidade Federal de Viçosa - Minas Gerais, Brazil.

Comments and Suggestions for Authors

  • Manuscript is well designed, written and discussed; however, there are important aspects which must be corrected prior to its acceptance.
  • Authors claimed that they are optimizing the extraction of essential oils, however they are also using different drying temperatures of the raw material, so, authors are optimizing two aspects, and therefore it must by shown in the title, introduction and objectives.

R- We appreciate the suggestion. As suggested, the aspects are now show in the title, abstract and o introduction. The changes are found in lines 2-5; 16-18 and 69-74.

  • This referee does not understand the the objective of the toxicity assay. Clarify this aspect if you consider it relevant on the research, if not eliminate please.

R- The essential oil of Ocimum basilicum is a bioactive with a proven insecticidal action. Its insecticidal potential has already been studied and proven for different insects pests of stored grain, which made it considered an important and promising alternative to synthetic insecticides. In this sense, the essential oil toxicity test against Sitophilus zeamais is of fundamental importance in optimizing the extraction process of this essential oil to identify which conditions of extraction best demonstrate its toxicological efficacy.

  • Line 151: there is a reference not properly presented for the Molecules format.

R-Thanks for the correction, the change was made and can be found in reference number 25.

  • Please use always same way of presentation of GC techniques, GC-MS and GC-FID, do not use GC/FID.

R- Thanks for the suggestion, the correction was made and can be found in lines: 160, 179, 258, 281 and 311.

-Authors must include another identification system by calculations of retention indexes (perhaps kovats) and compare them with the literature.

R-The authors appreciate the consideration. The comparison by the kovats index was carried out in our study, only it had not been mentioned in the text. This information is now present in the text. [lines 153-158 and 241-251].

Reviewer 2 Report

The manuscript discusses the optimization of the extraction method of essential oils of Basil using three factors: heating time for hydrodistillation, duration of ultrasound (at 100 °C) and drying temperature.

The work is very interesting, the methods used are extremely valid and the results are convincing.

There are problems in this manuscript.

The writing style is very limited. Furthermore, even if I am not a native English speaker, this manuscript requires proofreading and editing of English. It is also necessary, as is conventionally done in the articles, the use of past tense.

 The results obtained are very convincing for experts. However, these results must be accompanied by a study of life cycle analysis. Or simply by comparison with the "good old method of hydrodistillation".

Indeed, the best results are obtained by heating the most? My question, "a little provocative", what is the difference with hydrodistillation? and what energy cost? This is a central question in this work.

Methodological and mechanistic problem.

In the abstract L23-24. “The toxicity of the essential oil on S. zeamais increased with the use of ultrasound in its extraction". What the authors mean by this sentence? Please, explain. Are essential oils different? If it is the case, why?
In the discussion “P12 L323-P9-L325”. there is an "explanation".

 “This may be due to the use of ultrasound to allow an increase in the extraction of bioactive compounds and consequently to 324 improve the biological activity of the essential oils obtained.”

Is that possible?

According to the M&M “P5 L125-126.” Authors used "in 0.8 L glass flasks (8 cm in diameter x 15 cm in height) with 50 125 adults of non-sexed S. zeamais, in four replications. The concentration of essential oil used was 20 μL L-1 of air. "

If the concentration of EO is always the same, the explanation is not acceptable

Minor remarks

Results

P6 L191-192. Please, rewrite

L197-« were obtained in treatments 7, 12,14, 25 e 27 (Table 2) » Replace e by and.

P8 L216-2017 This sentence should be rewritted

References

Ref 7 Tetraclinis articulata in italic

Ref 40 Mejric instead of Mejri

This reference is false. The review was not found in Asian journal of Green Chemistry.

please modify

Reference 46. Please modify Journal of Essential Oil Bearing Plants . 2014, 17, 1075-1086

Figures

The quality of the figures is very low. Please submit sharper figures

Author Response

Viçosa, May 26, 2020

We are re-presenting the manuscript Manuscript ID molecules-798684 entitled "Optimization and validation of the Ocimum basilicum essential oil extraction method by the association of ultrasound and hydrodistillation techniques" with revision to be considered for publication at Molecules. We thank the editor and reviewers for their comments and suggestions as they have greatly improved the work. The modified sentences were marked in red and the English revisions in green.

Yours sincerely,

Prof. Dra. Lêda Rita D’Antonino Faroni

Department of Agricultural Engineering

Universidade Federal de Viçosa - Minas Gerais Brazil.

Comments and Suggestions for Authors

The manuscript discusses the optimization of the extraction method of essential oils of Basil using three factors: heating time for hydrodistillation, duration of ultrasound (at 100 °C) and drying temperature.The work is very interesting, the methods used are extremely valid and the results are convincing.There are problems in this manuscript.

-The writing style is very limited. Furthermore, even if I am not a native English speaker, this manuscript requires proofreading and editing of English. It is also necessary, as is conventionally done in the articles, the use of past tense.

R - The authors are thankful for the reviewer’s comment. The manuscript was thoroughly revised for English and scientific stylish writing.

- The results obtained are very convincing for experts. However, these results must be accompanied by a study of life cycle analysis. Or simply by comparison with the "good old method of hydrodistillation".

R- We appreciate the suggestion. Although we did not perform an extraction exclusively by the traditional method of hydrodistillation, this comparison was made through factorial planning by the extraction conditions presented in lines 9 and 26 of table 2, in which the sonification time was zero.

-Indeed, the best results are obtained by heating the most? My question, "a little provocative", what is the difference with hydrodistillation? and what energy cost? This is a central question in this work.

R- The best results were obtained when the leaves of Ocimum basilicum were subjected to lower drying temperatures, however in the extraction process all treatments were subjected to the same heating temperature (100 ° C). Hydrodistillation extraction, despite being the most widely used method, has some important deficiencies, such as the extended extraction time that can exceed 180 minutes (which results in higher costs and can induce the hydrolysis of some essential oil constituents) resulting in low yield and losses of volatile compounds due to prolonged heating. By optimizing the extraction process with the use of ultrasound and the best drying temperature of the raw material, the extraction time is significantly reduced and the volatile compounds are preserved, also causing a higher yield of the essential oil. Regarding the energy cost of the process, a study with this objective would be necessary, which was not done in our work.

-Methodological and mechanistic problem.

In the abstract L23-24. “ The toxicity of the essential oil on S. zeamais increased with the use of ultrasound in its extraction »What the authors mean by this sentence? Please, explain. Are essential oils different? If it is the case, why?
In the discussion “P12 L323-P9-L325”. there is an "explanation".

-  “This may be due to the use of ultrasound to allow an increase in the extraction of bioactive compounds and consequently to 324 improve the biological activity of the essential oils obtained.” Is that possible?

R- Only the essential oil of Ocimum basilicum was used in the extraction optimization process. The essential oil samples used are different with respect to the extraction conditions to which they were subjected (hydrodistillation time, sonication time and drying temperature). It was found that the samples of essential oil extracted with the use of ultrasound caused higher mortality in adults of S. zeamais when applied by fumigation. It is believed that the mechanical effect of ultrasound accelerates the release of organic compounds contained in the plant due to cell action, wall rupture and intensification of mass transfer, allowing for a better extraction of compounds that act on the toxicological activity of essential oil on insects.

- According to the M&M “P5 L125-126.” Authors used "in 0.8 L glass flasks (8 cm in diameter x 15 cm in height) with 50 125 adults of non-sexed S. zeamais, in four replications. The concentration of essential oil used was 20 μL L-1 of air. "

-If the concentration of EO is always the same, the explanation is not acceptable

R- The authors are thankful for the comment from the reviewer. Indeed the concentration of the essential oil was always the same (20 µL of essential oil per L air from glass flasks). However, the chemical composition of the oils are not always the same, given the different conditions of extractions. As discussed in the manuscript, different factors, like temperature, can have an impact on the chemical composition of essential oil. Thus, although the concentration of essential is the same, the chemical composition was affected. That’s why the application of the essential in the bioassay with insect it’s important to show that the bioactivity can vary according to the process used to extract the essential oil.

 Results

  • P6 L191-192. Please, rewrite

R -Thanks for the correction. The sentence was rewritten for more clarity. “The pareto graph with data of the yield of the essential oil of O. basilicum leaves in relation to dry matter when subjected to drying at different temperatures of the leaves, different hydrodistillation, and ultrasound times is shown in Figures 2, 3, and 4, respectively.” [line 195-197]

-L197-« were obtained in treatments 7, 12,14, 25 e 27 (Table 2) » Replace e by and.

R- Thanks for the correction. The due correction was made in the manuscript.        [line 201-204]

-P8 L216-217 This sentence should be rewritted

R- The authors are thankful for the suggestion. The sentence was rewritten as follows: “Results of the central composite planning show that the variables investigated (drying temperature, time of sonication by ultrasound and hydrodistillation time) were significant (p ≤ 0.05) for the toxicity of the essential oil of O. basilicum to S. zeamais (Figure 5).” [Line 221-223]

References

-Ref 7 Tetraclinis articulata in italic

R- Thanks for the correction. The due correction was made in the manuscript.

-Ref 40 Mejric instead of Mejri

R- Thanks for the correction. The due correction was made in the manuscript.

-This reference is false. The review was not found in Asian journal of Green Chemistry please modify

R- Thank you for the observation, but the reference is true, follow its doi for verification. Doi: 10.22034 / AJGC.2018.61443

-Reference 46. Please modify Journal of Essential Oil Bearing Plants2014, 17, 1075-1086

R- Thanks for the correction. The due correction was made in the manuscript.

Figures

-The quality of the figures is very low. Please submit sharper figures

R-Thanks for the correction. The due correction was made in the manuscript.

Round 2

Reviewer 1 Report

Manuscript has been largely improved.

Author Response

Viçosa, June 01, 2020

Dear editor and reviewer

We are re-presenting the manuscript Manuscript ID molecules-798684 entitled                "Optimization and validation of the Ocimum basilicum essential oil extraction method by the association of ultrasound and hydrodistillation techniques" with revision to be considered for publication at Molecules

The authors are thankful for the peer-review work carried out by the reviewer.

Yours sincerely,

 Prof. Dra. Lêda Rita D’Antonino Faroni

Department of Agricultural Engineering

Universidade Federal de Viçosa - Minas Gerais, Brazil.

 Reviewer 2 Report

The authors improved greatly the manuscript. Recommendations have been taken into account. Some convincing answers, dissipate the lack of clarity.

Nevertheless, others questions were revealed.

The majority of the recommendations have been considered. Some answers are convincing, but others have revealed other questions.
With the used extraction method, can we qualify the extract as  an essential oil?

according to the European Pharmacopoeia, only oils extracted with hydrodistillation are qualified as essential oils.

Depending on the authors' response, the temperature can influence the composition.

« As discussed in the manuscript, different factors, like temperature, can have an impact on the chemical composition of essential oil. Thus, although the concentration of essential is the same, the chemical composition was affected. »

1- is the extraction temperature? or the temperature of development of the plant?

2- have the authors considered the enantiomers that can be generated according to the method of extraction of essential oils (see the work of Uitterhaegen et al., 2018. Industrial Crops and Products, 122, 57-65. doi.org/10.1016/j.indcrop.2018.05.050)?

Could this be a possible explanation for this difference in activity?

This point remains unclear.

Other remarks

References are not rightly indexed.

Please ensure that all scientific names of plants and insects are in italic (Ref. 7, 17, 29, 37, 38,39, 41, 42, 47…)

Reference 7. Please replace Ind Cro Prod. by Ind. Crop. Prod.

Ref 16 replace Advances by Adv.

Ref 31 Replace African by Afr.

Ref 35 replace Comprehensive Reviews in Food Science and Food Safety by  Compr. Rev. Food Sci. Food Saf.

Ref 40 replace Rev. Bras. Eng. Agrícola e Ambient. By Rev. Bras. Eng. Agrí. Amb.

Ref 41 replace  Scientific reports by Sci. Rep.

Ref 42 replace Journal of Stored Products Research by

Ref 43. This was already requested in Versio 1 Please replace J. Essent by J. Essent. Oil Bear. Plant (see website : https://www.tandfonline.com/doi/abs/10.1080/0972060X.2014.935091)

Ref 44 Review Undergr Researc Agricl Life Scienc by Rev. Undergr Res. Agri. Life Sci.

Ref 50 Please provide the correct reference. As presented this reference did not exist in the website of Asian J. Green Chem.

Author Response

Viçosa, June 01, 2020

We are re-presenting the manuscript Manuscript ID molecules-798684 entitled "Optimization and validation of the Ocimum basilicum essential oil extraction method by the association of ultrasound and hydrodistillation techniques" with revision to be considered for publication at Molecules. We thank the editor and reviewers for their comments and suggestions as they have greatly improved the work. The modified sentences were marked in yellow.

Yours sincerely,

 Prof. Dra. Lêda Rita D’Antonino Faroni

Department of Agricultural Engineering

Universidade Federal de Viçosa - Minas Gerais, Brazil.

Comments and Suggestions for Authors

The authors improved greatly the manuscript. Recommendations have been taken into account. Some convincing answers, dissipate the lack of clarity.

Nevertheless, others questions were revealed.

The majority of the recommendations have been considered. Some answers are convincing, but others have revealed other questions.
With the used extraction method, can we qualify the extract as  an essential oil?

according to the European Pharmacopoeia, only oils extracted with hydrodistillation are qualified as essential oils.

R -The authors are thankful for the peer-review work carried out by the reviewer and for the comments and suggestions.

We partially agree with the reviewer that essential oil were traditionally extracted by hydrodistillation. However, there is a remarkable progress on studies on different extractions mode of essential oils. In this sense, essential oils are better charactherized as “composite mixtures of volatile compounds most frequently present at low concentrations in plants” rather than solely the extraction method used. Quoting Stratakos and Koidis (2016): “Several different extraction techniques are widely employed for the extraction of essential oils such as steam distillation and solvent extraction. These methods are characterized by drawbacks such as low extraction efficiency and selectivity, use of large amounts of solvents, and long extraction times. In many cases, the quality of the essential oil obtained by conventional methods can be influenced by hydrolyzation or oxidation than can take place due to long extraction time and/or high water quantity. Due to these limitations, alternative methods for the extraction of essential oils have been developed which can typically overcome these problems. Supercritical fluid extraction, microwave assisted extraction and ultrasound are novel methods that are now recognized as efficient extraction methods and can significantly reduce extraction times, enhance yields, and quality of essential oil.”. other authos do also recognize other methods for the extraction of essential oil, not only hidrodistilation. Quoting Aziz et al. (2018): “Several advanced (supercritical fluid extraction, subcritical extraction liquid, solvent-free microwave extraction) and conventional (hydrodistillation, steam distillation, hydrodiffusion, solvent extraction) methods have been discussed for the extraction of essential oils. Advanced methods are considered as the most promising extraction techniques due to less extraction time, low energy consumption, low solvent used and less carbon dioxide emission.”.

Aziz, Zarith AA, et al. "Essential oils: extraction techniques, pharmaceutical and therapeutic potential-a review." Current drug metabolism 19.13 (2018): 1100-1110. https://doi.org/10.2174/1389200219666180723144850

Stratakos, Alexandros Ch, and Anastasios Koidis. "Methods for extracting essential oils." Essential oils in food preservation, flavor and safety. Academic Press, 2016. 31-38.

https://doi.org/10.1016/B978-0-12-416641-7.00004-3

Sankarikutty, B., and C. S. Narayanan. "ESSENTIAL OILS| Isolation and Production." (2003): 2185-2189. https://doi.org/10.1016/B0-12-227055-X/00426-0

Depending on the authors' response, the temperature can influence the composition.

« As discussed in the manuscript, different factors, like temperature, can have an impact on the chemical composition of essential oil. Thus, although the concentration of essential is the same, the chemical composition was affected. »

  • is the extraction temperature? or the temperature of development of the plant?

R- We appreciate the comment. That is the drying temperature of the leaves before the extraction of essential oil.

2- have the authors considered the enantiomers that can be generated according to the method of extraction of essential oils (see the work of Uitterhaegen et al., 2018. Industrial Crops and Products, 122, 57-65. doi.org/10.1016/j.indcrop.2018.05.050)?

Could this be a possible explanation for this difference in activity?

This point remains unclear.

R- The authors are thankful for the reviewer bringing up this information. Unfortunately, the enantiomeric ratio was not evaluated in the present study. However, the authors do agree with the reviewer that this could be a possible explanation for the differences on biological activity. The authors have added this topic to discussion in lines 340-343 to reflect this possibility.

Other remarks

References are not rightly indexed.

Please ensure that all scientific names of plants and insects are in italic (Ref. 7, 17, 29, 37, 38,39, 41, 42, 47…)

R- The authors are thankful for the correction. The due correction was done in the manuscript.

Reference 7. Please replace Ind Cro Prod. by Ind. Crop. Prod.

R- Thank you. The replacement was done in the manuscript.

Ref 16 replace Advances by Adv.

R- Thank you. The replacement was done in the manuscript.

Ref 31 Replace African by Afr.

R - Thank you. The replacement was done in the manuscript.

Ref 35 replace Comprehensive Reviews in Food Science and Food Safety by  Compr. Rev. FoodSci. Food Saf.

R- The authors appreciate the correction. The replacement was done in the manuscript.

Ref 40 replace Rev. Bras. Eng. Agrícola e Ambient. By Rev. Bras. Eng. Agrí. Amb.

R- The replacement was done in the manuscript.

Ref 41 replace  Scientific reports by Sci. Rep.

R- Thank you. The replacement was done in the manuscript.

Ref 42 replace Journal of Stored Products Research by

R- The authors appreciate the correction. The replacement was done in the manuscript.

Ref 43. This was already requested in Versio 1 Please replace J. Essent by J. Essent. Oil Bear. Plant (see website : https://www.tandfonline.com/doi/abs/10.1080/0972060X.2014.935091)

R- The authors appreciate the correction. The replacement was done in the manuscript.

Ref 44 Review Undergr Researc Agricl Life Scienc by Rev. Undergr Res. Agri. Life Sci.

R- The authors appreciate the correction. The replacement was done in the manuscript.

Ref 50 Please provide the correct reference. As presented this reference did not exist in the website of Asian J. Green Chem.

R -The authors are greatful for the correction. The year of the publication was incorrect and it’s corrected now. The paper can be found on: http://www.ajgreenchem.com/article_61443.html. DOI of the publication: 10.22034/AJGC.2018.61443.
